# Generalization Bound and Learning Methods for Data-Driven Projections in Linear Programming

**Shinsaku Sakaue**
The University of Tokyo and RIKEN AIP
Tokyo, Japan
sakaue@mist.i.u-tokyo.ac.jp

**Taihei Oki**
Hokkaido University
Hokkaido, Japan
oki@icredd.hokudai.ac.jp

## Abstract

How to solve high-dimensional linear programs (LPs) efficiently is a fundamental question. Recently, there has been a surge of interest in reducing LP sizes using *random projections*, which can accelerate solving LPs independently of improving LP solvers. This paper explores a new direction of *data-driven projections*, which use projection matrices learned from data instead of random projection matrices. Given training data of $n$-dimensional LPs, we learn an $n \times k$ projection matrix with $n > k$. When addressing a future LP instance, we reduce its dimensionality from $n$ to $k$ via the learned projection matrix, solve the resulting LP to obtain a $k$-dimensional solution, and apply the learned matrix to it to recover an $n$-dimensional solution. On the theoretical side, a natural question is: how much data is sufficient to ensure the quality of recovered solutions? We address this question based on the framework of *data-driven algorithm design*, which connects the amount of data sufficient for establishing generalization bounds to the *pseudo-dimension* of performance metrics. We obtain an $\tilde{O}(nk^2)$ upper bound on the pseudo-dimension, where $\tilde{O}$ compresses logarithmic factors. We also provide an $\Omega(nk)$ lower bound, implying our result is tight up to an $\tilde{O}(k)$ factor. On the practical side, we explore two simple methods for learning projection matrices: PCA- and gradient-based methods. While the former is relatively efficient, the latter can sometimes achieve better solution quality. Experiments demonstrate that learning projection matrices from data is indeed beneficial: it leads to significantly higher solution quality than the existing random projection while greatly reducing the time for solving LPs.

## 1 Introduction

Linear programming (LP) has been one of the most fundamental tools used in various industrial domains [23, 18], and how to address high-dimensional LPs efficiently has been a major research subject in operations research. To date, researchers have developed various fast LP solvers, most of which stem from the simplex or interior-point method. Recent advances include a parallelized simplex method [27] and a randomized interior-point method [16]. Besides the improvements in LP solvers, there has been a growing interest in reducing LP sizes via *random projections* [44, 37, 2], motivated by the success of random sketching in numerical linear algebra [48]. Such a projection-based approach is *solver-agnostic* in that it can work with any solvers, including the aforementioned recent solvers, for solving reduced-size LPs. This solver-agnostic nature is beneficial, especially considering that LP solvers have evolved in distinct directions of simplex and interior-point methods.

In the context of numerical linear algebra, there has been a notable shift towards learning sketching matrices from data, instead of using random matrices [28, 29, 12, 33, 40]. This data-driven approach is effective when we frequently address similar instances. The line of previous research has demonstrated that learned sketching matrices can greatly improve the performance of sketching-based methods.

38th Conference on Neural Information Processing Systems (NeurIPS 2024).

## 1.1 Our contribution

Drawing inspiration from this background, we study a *data-driven projection* approach for accelerating repetitive solving of similar LP instances, which often arise in practice [21] (see also Remark 3.2). Our approach inherits the solver-agnostic nature of random projections for LPs, and it can improve solution quality by learning projection matrices from past LP instances. Our contribution is a cohesive study of this data-driven approach to LPs from both theoretical and practical perspectives, as follows.

**Generalization bound.** We first formulate the task of learning projection matrices as a statistical learning problem and study the generalization bound. Specifically, we analyze the number of LP-instance samples sufficient for bounding the gap between the empirical and expected objective values attained by the data-driven projection approach. Such a generalization bound is known to depend on the *pseudo-dimension* of the class of performance metrics. We prove an $\tilde{\mathrm{O}}(nk^2)$ upper bound on the pseudo-dimension (Theorem 4.4), where $n$ and $k$ are the original and reduced dimensionalities, respectively, and $\tilde{\mathrm{O}}$ compresses logarithmic factors. A main technical non-triviality lies in Lemma 4.3, which elucidates a piecewise polynomial structure of the optimal value of LPs as a function of input parameters. Besides playing a key role in proving Theorem 4.4, Lemma 4.3 offers general insight into the optimal value of LPs, which could have broader implications. We also give an $\Omega(nk)$ lower bound on the pseudo-dimension (Theorem 4.5). As experiments demonstrate later, we can get high-quality solutions with $k$ much smaller than $n$, suggesting our result, with only an $\tilde{\mathrm{O}}(k)$ gap, is nearly tight.

**Learning methods.** We then explore how to learn projection matrices in Section 5. We consider two simple learning methods based on principal component analysis (PCA) and gradient updates. The former efficiently constructs a projection matrix by extracting the top-$k$ subspace around which optimal solutions of future instances are expected to appear. The latter, although more costly, directly improves the optimal value of LPs via gradient ascent. In Section 6, experiments on various datasets confirm that projection matrices learned by the PCA- and gradient-based methods can lead to much higher solution quality than random projection [2], while greatly reducing the time for solving LPs.

## 1.2 Related work

**Random projections for LPs.** Vu et al. [44] introduced a random-projection method to reduce the number of equality constraints, and Poirion et al. [37] extended it to inequality constraints. As discussed therein, reducing the number of inequality constraints of LPs corresponds to reducing the dimensionality (the number of variables) of dual LPs. Recently, Akchen and Mišić [2] developed a column-randomized method for reducing the dimensionality of LPs. While these studies provide high probability guarantees, we focus on data-driven projections and discuss generalization bounds.

**Data-driven algorithm design.** *Data-driven algorithm design* [7], initiated by Gupta and Roughgarden [25], has served as a foundational framework for analyzing generalization bounds of various data-driven algorithms [8, 10, 11, 12, 39, 9]. Our statistical learning formulation in Section 3 and a general proof idea in Section 4 are based on it. Among the line of studies, the analysis technique for data-driven integer-programming (IP) methods [10, 11] is close to ours. The difference is that while their technique is intended for analyzing IP methods (particularly, branch-and-cut methods), we focus on LPs and discuss a general property of the optimal value viewed as a function of input parameters. Thus, our analysis is independent of solution methods, unlike the previous studies. This aspect is crucial for analyzing our solver-agnostic approach. Some studies have also combined LP/IP methods with machine learning [14, 21, 41], while learning of projection matrices has yet to be studied.

**Learning through optimization.** Our work is also related to the broad stream of research on learning through optimization procedures [4, 47, 1, 13, 42, 36, 3, 20, 45, 19], which we discuss in Appendix A.

**Notation.** For a positive integer $n$, let $\mathbf{I}_n$ and $\mathbf{0}_n$ be the $n \times n$ identity matrix and the $n$-dimensional all-zero vector, respectively, where we omit the subscript when it is clear from the context. For two matrices $\boldsymbol{A}$ and $\boldsymbol{B}$ with the same number of rows (columns), $[\boldsymbol{A}, \boldsymbol{B}]$ ($[\boldsymbol{A}; \boldsymbol{B}]$) denotes the matrix obtained by horizontally (vertically) concatenating $\boldsymbol{A}$ and $\boldsymbol{B}$.

## 2 Reducing dimensionality of LPs via projection

We overview the projection-based approach for reducing the dimensionality of LPs [37, 2]. For ease of dealing with feasibility issues, we focus on the following inequality-form LP with input parameters

$c \in \mathbb{R}^n$, $A \in \mathbb{R}^{m \times n}$, and $b \in \mathbb{R}^m$:

$$\text{maximize}_{x \in \mathbb{R}^n} \quad c^\top x \qquad \text{subject to} \quad Ax \leq b. \tag{1}$$

When $n$ is large, restricting variables to a low-dimensional subspace can be helpful for computing an approximate solution to (1) quickly. Specifically, given a *projection matrix* $P \in \mathbb{R}^{n \times k}$ with $n > k$, we consider solving the following *projected LP*, instead of (1):

$$\text{maximize}_{y \in \mathbb{R}^k} \quad c^\top P y \qquad \text{subject to} \quad AP y \leq b. \tag{2}$$

Once we get an optimal solution $y^*$ to the projected LP (2), we can recover an $n$-dimensional solution, $\tilde{x} = Py^*$, to the original LP (1). Note that the recovered solution is always feasible for (1), although not always optimal. We measure the solution quality with the objective value $c^\top \tilde{x} = c^\top Py^*$. Ideally, if $P$'s columns span a linear subspace that contains an optimal solution to (1), the recovered solution $\tilde{x} = Py^*$ is optimal to (1) due to the optimality of $y^*$ to (2). Therefore, if we find such a good $P$ close to being ideal with small $k$, we can efficiently obtain a high-quality solution $\tilde{x} = Py^*$ to (1) by solving the smaller projected LP (2).

**Remark 2.1** (Solver-specific aspects). As mentioned in Section 1, this projection-based approach is solver-agnostic in that we can apply any LP solver to projected LPs (2). To preserve this nature, we focus on designing projection matrices and do not delve into solver-specific discussions. Experiments in Section 6 will use Gurobi as a fixed LP solver, which is a standard choice. Strictly speaking, projections alter the sparsity and numerical stability of projected LPs, which can affect the performance of solvers. This point can be important, especially when original LPs are sparse and solvers exploit the sparsity. Investigating how to take such solver-specific aspects into account is left for future work.

## 3 Data-driven projection

While the previous studies [44, 37, 2] have reduced LP sizes via random projections, we may be able to improve solution quality by learning projection matrices from data. We formalize this idea as a statistical learning problem. Let $\Pi$ denote the set of all possible LP instances and $\mathcal{D}$ an unknown distribution on $\Pi$. Given LP instances sampled from $\mathcal{D}$, our goal is to learn $P$ that maximizes the expected optimal value of projected LPs over $\mathcal{D}$. Below, we assume the following three conditions.

**Assumption 3.1.** *(i) Every $\pi \in \Pi$ takes the inequality form (1), (ii) $x = 0_n$ is feasible for all $\pi \in \Pi$, and (iii) optimal values of all instances in $\Pi$ are upper bounded by a finite constant $H > 0$.*

Although Assumption 3.1 narrows the class of LPs we can handle, it is not as restrictive as it seems. Suppose for example that LP instances in $\Pi$ have identical equality constraints. While such LPs in their current form do not satisfy (i), we can convert them into the inequality form (1) by considering the null space of the equality constraints (see Appendix C for details), hence satisfying (i). This conversion is useful for dealing with LPs of maximum-flow and minimum-cost-flow problems on a fixed graph topology, where we can remove the flow-conservation equality constraints by considering the null space of the incidence matrix of the graph. Regarding condition (ii), we may instead assume that there exists an arbitrary common feasible solution $x_0$ without loss of generality. This is because we can translate the feasible region so that $x_0$ coincides with the origin $0_n$. Condition (ii) also implies that for any $P \in \mathbb{R}^{n \times k}$, projected LPs are *feasible* (i.e., their feasible regions are non-empty) since $y = 0_k$ is always feasible for any projected LPs. Condition (iii) is satisfied simply by focusing on *bounded* LPs (i.e., LPs with finite optimal values) and setting $H$ to the largest possible optimal value. Conditions (ii) and (iii) also ensure that the optimal value of projected LPs always lies in $[0, H]$, which is used to derive a generalization bound in Section 4. In Section 6, we see that many problems, including packing and network flow problems, can be written as LPs satisfying Assumption 3.1.

Due to condition (i), we can identify each LP instance $\pi \in \Pi$ with its input parameters $(c, A, b)$ in (1), i.e., $\pi = (c, A, b)$. For an LP instance $\pi \in \Pi$ and a projection matrix $P \in \mathbb{R}^{n \times k}$, we define

$$u(P, \pi) = \max\{ c^\top Py \ : \ AP y \leq b \} \tag{3}$$

as the optimal value of the projected LP. Our goal is to learn $P \in \mathbb{R}^{n \times k}$ from LP instances sampled from $\mathcal{D}$ to maximize the expected optimal value on future instances, i.e., $\mathbb{E}_{\pi \sim \mathcal{D}}[u(P, \pi)]$.

**Remark 3.2** (Validity of the setting). The above statistical learning setting regarding LP instances is not an artifact. As Fan et al. [21] discussed, LPs often serve as descriptive models, and each instance can be viewed as a realization of input parameters following some distribution. Such a scenario arises in, for example, daily production planning and flight scheduling. Note that the statistical learning setting is also widely used as a foundational framework in data-driven algorithm design [25, 8, 12, 9].

# 4 Generalization bound

This section studies the generalization bound, namely, how many samples from $\mathcal{D}$ are sufficient for guaranteeing that the expected optimal value, $\mathbb{E}_{\pi \sim \mathcal{D}}[u(\boldsymbol{P}, \pi)]$, of learned $\boldsymbol{P}$ is close to the empirical optimal value on sampled instances. First, let us overview the basics of learning theory. Let $\mathcal{U} \subseteq \mathbb{R}^{\Pi}$ be a class of functions, where each $u \in \mathcal{U}$ takes some input $\pi \in \Pi$ and returns a real value. We use the following *pseudo-dimension* [38] to measure the complexity of a class of real-valued functions.

**Definition 4.1.** Let $N$ be a positive integer. We say $\mathcal{U} \subseteq \mathbb{R}^{\Pi}$ *shatters* an input set, $\{\pi_1, \ldots, \pi_N\} \subseteq \Pi$, if there exist threshold values, $t_1, \ldots, t_N \in \mathbb{R}$, such that each of all the $2^N$ outcomes of $\{ u(\pi_i) \geq t_i : i = 1, \ldots, N \}$ is realized by some $u \in \mathcal{U}$. The *pseudo-dimension* of $\mathcal{U}$, denoted by $\mathrm{pdim}(\mathcal{U})$, is the maximum size of an input set that $\mathcal{U}$ can shatter.

In our case, the set $\mathcal{U}$ consists of functions $u(\boldsymbol{P}, \cdot) : \Pi \to \mathbb{R}$, defined in (3), for all possible projection matrices $\boldsymbol{P} \in \mathbb{R}^{n \times k}$. Each $u(\boldsymbol{P}, \cdot) \in \mathcal{U}$ takes an LP instance $\pi = (\boldsymbol{c}, \boldsymbol{A}, \boldsymbol{b}) \in \Pi$ as input and returns the optimal value of the projected LP. Assumption 3.1 ensures that the range of $u(\boldsymbol{P}, \cdot)$ is bounded by $[0, H]$ for all $\boldsymbol{P} \in \mathbb{R}^{n \times k}$. Thus, the well-known uniform convergence result (see, e.g., Anthony and Bartlett [5, Theorem 19.2] and Balcan [7, Theorem 29.2]) implies that for any distribution $\mathcal{D}$ on $\Pi$, $\varepsilon > 0$, and $\delta \in (0, 1)$, if $N = \Omega((H/\varepsilon)^2(\mathrm{pdim}(\mathcal{U}) + \log(1/\delta)))$ instances drawn i.i.d. from $\mathcal{D}$ are given, with probability at least $1 - \delta$, for all $\boldsymbol{P} \in \mathbb{R}^{n \times k}$, it holds that

$$\left| \frac{1}{N} \sum_{i=1}^{N} u(\boldsymbol{P}, \pi_i) - \mathbb{E}_{\pi \sim \mathcal{D}}[u(\boldsymbol{P}, \pi)] \right| \leq \varepsilon. \tag{4}$$

That is, if a projection matrix $\boldsymbol{P}$ produces high-quality solutions on $N \approx (H/\varepsilon)^2 \cdot \mathrm{pdim}(\mathcal{U})$ instances sampled i.i.d. from $\mathcal{D}$, it likely yields high-quality solutions on future instances from $\mathcal{D}$ as well. Thus, analyzing $\mathrm{pdim}(\mathcal{U})$ of $\mathcal{U} = \{ u(\boldsymbol{P}, \cdot) : \Pi \to \mathbb{R} : \boldsymbol{P} \in \mathbb{R}^{n \times k} \}$ reveals the sufficient sample size.

**Remark 4.2** (Importance of uniform convergence). While the above generalization bound is not the sole focus of learning theory, it is particularly valuable in data-driven algorithm design, as is also discussed in the literature [8, 10]. Note that (4) holds uniformly for all $\boldsymbol{P} \in \mathbb{R}^{n \times k}$, offering performance guarantees *regardless of how $\boldsymbol{P}$ is learned*. Thus, we may select learning methods based on their empirical performance. This is helpful since there are no gold-standard methods for learning parameters of algorithms; we discuss learning methods for our case in Section 5. This situation differs from the standard supervised learning setting, where we minimize common losses, e.g., squared and logistic. Additionally, the uniform bound ensures that learned $\boldsymbol{P}$ does not overfit sampled instances.

## 4.1 Upper bound on $\mathrm{pdim}(\mathcal{U})$

Building upon the above learning theory background, a crucial factor for establishing the generalization bound is $\mathrm{pdim}(\mathcal{U})$. To upper bound this, we give a structural observation of the optimal value of LPs (Lemma 4.3) and combine it with a general proof idea in data-driven algorithm design [25, 7].

We first overview the general proof idea. Suppose that we have an upper bound on the number of outcomes of $\{ u(\boldsymbol{P}, \pi_i) \geq t_i : i = 1, \ldots, N \}$ that grows more slowly than $2^N$. Since shattering $N$ instances requires $2^N$ outcomes, the largest $N$, such that the upper bound is at least $2^N$, serves as an upper bound on $\mathrm{pdim}(\mathcal{U})$ (intuitively, $\mathrm{pdim}(\mathcal{U}) \lesssim \log_2($"upper bound on the number of outcomes"$)$). Below, we discuss bounding the number of outcomes, which is the most technically important step.

To examine the number of possible outcomes, we consider a fundamental question related to sensitivity analysis of LPs: *how does the optimal value of an LP behave when input parameters change?*[1] In our case, a projected LP has input parameters $(\boldsymbol{P}^\top \boldsymbol{c}, \boldsymbol{A}\boldsymbol{P}, \boldsymbol{b}) \in \mathbb{R}^k \times \mathbb{R}^{m \times k} \times \mathbb{R}^m$, where $\boldsymbol{P}^\top \boldsymbol{c}$ and $\boldsymbol{A}\boldsymbol{P}$ change with $\boldsymbol{P} \in \mathbb{R}^{n \times k}$. Thus, addressing this question offers insight into the number of outcomes. Lemma 4.3 provides an answer for a more general setting, which might find other applications beyond our case since learning through LPs is not limited to the projection-based approach [47, 13, 42, 20].

**Lemma 4.3.** *Let $t \in \mathbb{R}$ be a threshold value. Consider an LP $\tilde{\pi} = (\tilde{\boldsymbol{c}}, \tilde{\boldsymbol{A}}, \tilde{\boldsymbol{b}}) \in \mathbb{R}^k \times \mathbb{R}^{m \times k} \times \mathbb{R}^m$ such that each entry of $\tilde{\boldsymbol{c}}$, $\tilde{\boldsymbol{A}}$, and $\tilde{\boldsymbol{b}}$ is a polynomial of degree at most $d$ in $\nu$ real variables, $\boldsymbol{\theta} \in \mathbb{R}^\nu$.*

---

[1]While a similar question is studied in Balcan et al. [11, Theorem 3.1], their result focuses on the case where new constraints are added to LPs to analyze branch-and-cut methods, unlike our Lemma 4.3. In our case, we need to care about rank-deficient input matrices, which we circumvent via the reformulation given at the beginning of the proof of Lemma 4.3.

*Assume that $\tilde{\pi}$ is bounded and feasible for every $\boldsymbol{\theta} \in \mathbb{R}^\nu$. Then, there are up to $\binom{m+2k}{2k}(m+2k+2)$ polynomials of degree at most $(2k+1)d$ in $\boldsymbol{\theta}$ whose sign patterns ($<0$, $=0$, or $>0$) partition $\mathbb{R}^\nu$ into some regions, and whether $\max\{\tilde{\boldsymbol{c}}^\top \boldsymbol{y} : \tilde{\boldsymbol{A}}\boldsymbol{y} \le \tilde{\boldsymbol{b}}\} \ge t$ or not is identical within each region.*

*Proof.* First, we rewrite the LP $\tilde{\pi} = (\tilde{\boldsymbol{c}}, \tilde{\boldsymbol{A}}, \tilde{\boldsymbol{b}})$ as an equivalent $2k$-dimensional LP with non-negativity constraints: $\max\{\tilde{\boldsymbol{c}}^\top(\boldsymbol{y}^+ - \boldsymbol{y}^-) : \tilde{\boldsymbol{A}}(\boldsymbol{y}^+ - \boldsymbol{y}^-) \le \tilde{\boldsymbol{b}}, [\boldsymbol{y}^+; \boldsymbol{y}^-] \ge \boldsymbol{0}\}$. The resulting constraint matrix, $\boldsymbol{A}' := [\tilde{\boldsymbol{A}}, -\tilde{\boldsymbol{A}}; -\boldsymbol{I}_{2k}]$, has full column rank, which simplifies the subsequent discussion. Note that the maximum degree of input parameters remains at most $d$, while the sizes, $m$ and $k$, increase to $m' := m + 2k$ and $k' := 2k$, respectively. Below, we focus on the reformulated LP $(\boldsymbol{c}', \boldsymbol{A}', \boldsymbol{b}') \in \mathbb{R}^{k'} \times \mathbb{R}^{m' \times k'} \times \mathbb{R}^{m'}$, where $\boldsymbol{c}' := [\tilde{\boldsymbol{c}}; -\tilde{\boldsymbol{c}}]$, $\boldsymbol{b}' := [\tilde{\boldsymbol{b}}; \boldsymbol{0}_{2k}]$, and $\boldsymbol{A}'$ has full column rank.

We consider determining $\max\{\boldsymbol{c}'^\top \boldsymbol{y} : \boldsymbol{A}'\boldsymbol{y} \le \boldsymbol{b}'\} \ge t$ or not by checking all vertices of the feasible region. For any size-$k'$ subset, $I \subseteq \{1, \ldots, m'\}$, of row indices of $\boldsymbol{A}' \in \mathbb{R}^{m' \times k'}$, let $\boldsymbol{A}'_I$ denote the $k' \times k'$ submatrix of $\boldsymbol{A}'$ with rows restricted to $I$ and $\boldsymbol{b}'_I \in \mathbb{R}^{k'}$ the corresponding subvector of $\boldsymbol{b}'$. For every subset $I$ with $\det \boldsymbol{A}'_I \ne 0$, let $\boldsymbol{y}_I := \boldsymbol{A}'_I^{-1}\boldsymbol{b}'_I$. Since the LP is bounded and feasible, and $\boldsymbol{A}'$ has full column rank, there is a vertex optimal solution written as $\boldsymbol{y}_I = \boldsymbol{A}'_I^{-1}\boldsymbol{b}'_I$ for some $I$ (see the proof of Korte and Vygen [31, Proposition 3.1]). Thus, the optimal value is at least $t$ if and only if there exists at least one size-$k'$ subset $I$ with $\det \boldsymbol{A}'_I \ne 0$, $\boldsymbol{A}'\boldsymbol{y}_I \le \boldsymbol{b}'$, and $\boldsymbol{c}'^\top \boldsymbol{y}_I \ge t$.

Based on the above observation, we identify polynomials whose sign patterns determine $\max\{\boldsymbol{c}'^\top \boldsymbol{y} : \boldsymbol{A}'\boldsymbol{y} \le \boldsymbol{b}'\} \ge t$ or not. For any subset $I$, if $\det \boldsymbol{A}'_I \ne 0$, Cramer's rule implies that $\boldsymbol{y}_I = \boldsymbol{A}'_I^{-1}\boldsymbol{b}'_I$ is written as $\boldsymbol{f}_I(\boldsymbol{\theta})/\det \boldsymbol{A}'_I$, where $\boldsymbol{f}_I(\boldsymbol{\theta})$ is some $k'$-valued polynomial vector of $\boldsymbol{\theta}$ with degrees at most $k'd$. Thus, we can check $\boldsymbol{A}'\boldsymbol{y}_I \le \boldsymbol{b}'$ and $\boldsymbol{c}'^\top \boldsymbol{y}_I \ge t$ by examining sign patterns of $m' + 1$ polynomials, $\boldsymbol{A}'\boldsymbol{f}_I(\boldsymbol{\theta}) - (\det \boldsymbol{A}'_I)\boldsymbol{b}'$ and $\boldsymbol{c}'^\top \boldsymbol{f}_I(\boldsymbol{\theta}) - t\det \boldsymbol{A}'_I$, whose degrees are at most $(k'+1)d$. Considering all the $\binom{m'}{k'}$ choices of $I$, there are $\binom{m'}{k'}(m'+2)$ polynomials of the form $\det \boldsymbol{A}'_I$, $\boldsymbol{A}'\boldsymbol{f}_I(\boldsymbol{\theta}) - (\det \boldsymbol{A}'_I)\boldsymbol{b}'$, and $\boldsymbol{c}'^\top \boldsymbol{f}_I(\boldsymbol{\theta}) - t\det \boldsymbol{A}'_I$ with degrees at most $(k'+1)d$ such that their sign patterns partition $\mathbb{R}^\nu$ into some regions, and $\max\{\boldsymbol{c}'^\top \boldsymbol{y} : \boldsymbol{A}'\boldsymbol{y} \le \boldsymbol{b}'\} \ge t$ or not is identical within each region. Substituting $m + 2k$ and $2k$ into $m'$ and $k'$, respectively, completes the proof. $\square$

Lemma 4.3 states that the outcome of whether $u(\boldsymbol{P}, \pi) = \max\{\boldsymbol{c}^\top \boldsymbol{P}\boldsymbol{y} : \boldsymbol{A}\boldsymbol{P}\boldsymbol{y} \le \boldsymbol{b}\}$ exceeds $t$ or not is determined by sign patterns of polynomials of $\boldsymbol{P}$, and an upper bound on the sign patterns of polynomials is known as *Warren's theorem* [46], as detailed shortly. Combining them with the aforementioned general idea yields the following upper bound on $\mathrm{pdim}(\mathcal{U})$.

**Theorem 4.4.** $\mathrm{pdim}(\mathcal{U}) = \mathrm{O}(nk^2 \log mk)$.

*Proof.* Let $(\pi, t) \in \Pi \times \mathbb{R}$ be a pair of an LP instance and a threshold value. Setting $\boldsymbol{\theta} = \boldsymbol{P}$ and $d = 1$ in Lemma 4.3, we have up to $\binom{m+2k}{k}(m+2k+2)$ polynomials of degree at most $2k+1$ whose sign patterns determine whether $u(\boldsymbol{P}, \pi) \ge t$ or not. Thus, given $N$ pairs of input instances and threshold values, $(\pi_i, t_i)_{i=1}^N$, we have up to $N \times \binom{m+2k}{2k}(m+2k+2)$ polynomials whose sign patterns determine $u(\boldsymbol{P}, \pi_i) \ge t_i$ or not for all $i = 1, \ldots, N$, i.e., outcomes of $N$ instances.

Warren's theorem states that given $\ell$ polynomials of $\nu$ variables with degrees at most $\Delta$, the number of all possible sign patterns is at most $(8e\ell\Delta/\nu)^\nu$ [46] (see also Goldberg and Jerrum [24, Corollary 2.1]). In our case, the number of polynomials is $\ell = N \times \binom{m+2k}{2k}(m+2k+2)$, and each of them has $\nu = nk$ variables ($\boldsymbol{P}$'s entries) and degrees at most $\Delta = 2k+1$. Thus, the number of all possible outcomes is at most $\left(8eN\binom{m+2k}{2k}\frac{(m+2k+2)(2k+1)}{nk}\right)^{nk} \lesssim \left(\frac{N}{nk}\right)^{nk}\mathrm{poly}(m, k)^{nk^2}$. To shatter the set of $N$ instances, the right-hand side must be at least $2^N$. Taking the base-2 logarithm, it must hold that $N \lesssim nk \log_2 \frac{N}{nk} + \mathrm{O}(nk^2 \log mk) \le \frac{2}{3}N + \mathrm{O}(nk^2 \log mk)$, where we used $x\log_2 \frac{1}{x} \le \frac{2}{3}$ for $x > 0$. Therefore, $\mathcal{U}$ can shatter $\mathrm{O}(nk^2 \log mk)$ instances, obtaining the desired bound on $\mathrm{pdim}(\mathcal{U})$. $\square$

### 4.2 Lower bound on $\mathrm{pdim}(\mathcal{U})$

We then provide an $\Omega(nk)$ lower bound on $\mathrm{pdim}(\mathcal{U})$ to complement the above $\tilde{\mathrm{O}}(nk^2)$ upper bound, implying the tightness up to an $\tilde{\mathrm{O}}(k)$ factor. See Appendix B for the proof.

**Theorem 4.5.** $\mathrm{pdim}(\mathcal{U}) = \Omega(nk)$.

Our proof indeed gives the same lower bound on the $\gamma$-*fat shattering dimension* for $\gamma < 1/2$, which implies a lower bound of $\Omega(nk/\varepsilon)$ on $N$, the sample size needed to guarantee (4) [5, Theorem 19.5]. Thus, in terms of the sample complexity, our result is tight up to an $\tilde{O}(k/\varepsilon)$ factor. The $1/\varepsilon$ gap is inevitable in general [5, Section 19.5], while closing the $\tilde{O}(k)$ gap is an interesting open problem.

## 5 Learning methods

We then discuss how to learn projection matrices from training datasets. From the bound (4), given $N$ training LP instances, the expected solution quality on future instances likely remains within the range of $\pm\varepsilon$ from the empirical one, where $\varepsilon \lesssim H\sqrt{\mathrm{pdim}(\mathcal{U})/N} \lesssim Hk\sqrt{n/N}$ due to Theorem 4.4, *regardless of how we learn a projection matrix $\boldsymbol{P}$*. Therefore, in practice, we only need to find an empirically good projection matrix $\boldsymbol{P}$, which motivates us to explore various ideas for learning $\boldsymbol{P}$. Below, we discuss two natural ideas: PCA- and gradient-based methods.

**Remark 5.1** (Training time). We emphasize that learning methods are used only before addressing future LP instances and not once a projection matrix $\boldsymbol{P}$ is learned. Hence, they can take much longer than the time for solving new LP instances. Similarly, we suppose that optimal solutions to training instances are available, as we can compute them a priori. Note that similar premises are common in most data-driven algorithm research [28, 14, 12, 9, 21, 41]. Considering this, our learning methods are primarily intended for conceptual simplicity, not for efficiency. For completeness, we present the theoretical time complexity and the training time taken in the experiments in Appendix E.

### 5.1 PCA-based method

As described in Section 2, a projection matrix $\boldsymbol{P}$ should preferably have columns that span a low-dimensional subspace around which future optimal solutions will appear. Hence, a natural idea is to use PCA to extract such a subspace, regarding optimal solutions to training instances as data points.

Formally, let $\boldsymbol{X} \in \mathbb{R}^{N\times n}$ be a matrix whose $i$th row is an optimal solution to the $i$th training instance. We apply PCA to this $\boldsymbol{X}$. Specifically, we subtract the mean, $\bar{\boldsymbol{x}} = \frac{1}{N}\boldsymbol{X}^\top\mathbf{1}_N$, from each row of $\boldsymbol{X}$ and apply the singular value decomposition (SVD) to $\boldsymbol{X} - \mathbf{1}_N\bar{\boldsymbol{x}}^\top$, obtaining a decomposition of the form $\boldsymbol{U}\boldsymbol{\Sigma}\boldsymbol{V}^\top = \boldsymbol{X} - \mathbf{1}_N\bar{\boldsymbol{x}}^\top$. Let $\boldsymbol{V}_{k-1} \in \mathbb{R}^{n\times(k-1)}$ be the submatrix of $\boldsymbol{V}$ whose columns are the top-$(k-1)$ right-singular vectors of $\boldsymbol{X} - \mathbf{1}_N\bar{\boldsymbol{x}}^\top$. We use $\boldsymbol{P} = [\bar{\boldsymbol{x}}, \boldsymbol{V}_{k-1}] \in \mathbb{R}^{n\times k}$ as a projection matrix. Here, $\bar{\boldsymbol{x}}$ is concatenated due to the following consideration: since $\boldsymbol{V}_{k-1}$ is designed to satisfy $\boldsymbol{V}_{k-1}\boldsymbol{Y}' \approx \boldsymbol{X}^\top - \bar{\boldsymbol{x}}\mathbf{1}_N^\top$ for some $\boldsymbol{Y}' \in \mathbb{R}^{(k-1)\times N}$, we expect $[\bar{\boldsymbol{x}}, \boldsymbol{V}_{k-1}]\boldsymbol{Y} \approx \boldsymbol{X}^\top$ to hold for some $\boldsymbol{Y} \in \mathbb{R}^{k\times N}$, hence $\boldsymbol{P} = [\bar{\boldsymbol{x}}, \boldsymbol{V}_{k-1}]$. This method is not so costly when optimal solutions to training LP instances are given, as it only requires finding the top-$(k-1)$ right-singular vectors of $\boldsymbol{X} - \mathbf{1}_N\bar{\boldsymbol{x}}^\top$.

### 5.2 Gradient-based method

While the PCA-based method aims to extract the subspace into which future optimal solutions are likely to fall, it only uses optimal solutions and discards input parameters of LPs. As a complementary approach, we provide a gradient-based method that directly improves the optimal value of LPs.

As a warm-up, consider maximizing $u(\boldsymbol{P}, \pi) = \max\{\boldsymbol{c}^\top\boldsymbol{P}\boldsymbol{y} : \boldsymbol{A}\boldsymbol{P}\boldsymbol{y} \le \boldsymbol{b}\}$ of a single LP instance $\pi = (\boldsymbol{c}, \boldsymbol{A}, \boldsymbol{b})$ via gradient ascent. Assume that the projected LP satisfies a *regularity condition*, which requires the existence of an optimal solution $\boldsymbol{y}^*$ at which active constraints are linearly independent. Then, $u(\boldsymbol{P}, \pi)$ is differentiable in $\boldsymbol{P}$ and the gradient is expressed as follows [42, Theorem 1] (see Appendix D for details of the derivation):

$$\nabla u(\boldsymbol{P}, \pi) = \boldsymbol{c}\boldsymbol{y}^{*\top} - \boldsymbol{A}^\top\boldsymbol{\lambda}^*\boldsymbol{y}^{*\top}, \tag{5}$$

where $\boldsymbol{\lambda}^* \in \mathbb{R}^m$ is a dual optimal solution. Thus, we can use the gradient ascent method to maximize $u(\boldsymbol{P}, \pi)$ under the regularity condition. However, this condition is sometimes prone to be violated, particularly when *Slater's condition* does not hold (i.e., there is no strictly feasible solution). For example, if the original LP has a constraint $\boldsymbol{x} \ge \mathbf{0}_n$ and every column of $\boldsymbol{P}$ has opposite-sign entries, it is likely that only $\boldsymbol{y} = \mathbf{0}_k$ satisfies $\boldsymbol{P}\boldsymbol{y} \ge \mathbf{0}_n$ by equality, which is the unique optimal solution but not strictly feasible. In this case, the regularity condition is violated since all rows of $\boldsymbol{P} \in \mathbb{R}^{n\times k}$ are active at $\mathbf{0}_k$ and linearly dependent due to $n > k$. To alleviate this issue, we apply the following

Table 1: Sizes of inequality-form LPs, where $m$ ($n$) represents the number of constraints (variables).

|  | Packing | MaxFlow | MinCostFlow | GROW7 | ISRAEL | SC205 | SCAGR25 | STAIR |
|---|---|---|---|---|---|---|---|---|
| $m$ | 50 | 1000 | 1000 | 581 | 316 | 317 | 671 | 696 |
| $n$ | 500 | 500 | 500 | 301 | 142 | 203 | 500 | 467 |

projection for $j = 1, \dots, k$ before computing the gradient in (5):

$$\boldsymbol{P}_{:,j} \leftarrow \arg\min_{\boldsymbol{x} \in \mathbb{R}^n} \{ \, \|\boldsymbol{x} - \boldsymbol{P}_{:,j}\|_2 \ : \ \boldsymbol{A}\boldsymbol{x} \leq \boldsymbol{b} \, \}, \tag{6}$$

where $\boldsymbol{P}_{:,j}$ denotes the $j$th column of $\boldsymbol{P}$. This minimally changes each column $\boldsymbol{P}_{:,j}$ to satisfy the original constraints. Consequently, any convex combination of $\boldsymbol{P}$'s columns is feasible for the original LP, increasing the chance that there exists a strictly feasible solution in $\{ \, \boldsymbol{y} \in \mathbb{R}^k \ : \ \boldsymbol{A}\boldsymbol{P}\boldsymbol{y} \leq \boldsymbol{b} \, \}$, although it is not guaranteed. This improves the likelihood that the regularity condition is satisfied.

Given $N$ training instances, $\pi_1, \dots, \pi_N$, we repeatedly update $\boldsymbol{P}$ as with SGD: for each $\pi_i$, we iterate to compute the gradient (5) and to update $\boldsymbol{P}$ with it. The projection (6) onto the feasible region of $\pi_i$ comes before computing the gradient for $\pi_i$. We call this method SGA (stochastic gradient ascent).

### 5.3 Final projection for feasibility

The previous discussion suggests that making each column of $\boldsymbol{P}$ feasible for training LP instances can increase the likelihood that future LP instances projected by $\boldsymbol{P}$ will have strictly feasible solutions. Considering this, after obtaining a projection matrix $\boldsymbol{P}$ with either the PCA- or gradient-based method, we project each column of $\boldsymbol{P}$ onto the intersection of the feasible regions of training LP instances, which we call the *final projection*. This can be done similarly to (6) replacing the constraints with $[\boldsymbol{A}_1; \dots; \boldsymbol{A}_N]\boldsymbol{x} \leq [\boldsymbol{b}_1; \dots; \boldsymbol{b}_N]$. If $\boldsymbol{A}_1, \dots, \boldsymbol{A}_N$ are identical, we can do it more efficiently by replacing the constraints with $\boldsymbol{A}_1 \boldsymbol{x} \leq \min\{\boldsymbol{b}_1, \dots, \boldsymbol{b}_N\}$, where the minimum is taken element-wise. Note that although the final projection can be costly for large $N$, we need to do it only once at the end of learning $\boldsymbol{P}$. This final projection never fails since $\boldsymbol{0}_n$ is always feasible as in Assumption 3.1.

## 6 Experiments

We experimentally evaluate the data-driven projection approach.[2] We used MacBook Air with Apple M2 chip, 24 GB of memory, and macOS Sonoma 14.1. We implemented algorithms in Python 3.9.7 with NumPy 1.23.2. We used Gurobi 10.0.1 [26] for solving LPs and computing projection in (6). We used the following three synthetic and five realistic datasets, each of which consists of 300 LP instances (200 for training and 100 for testing). Table 1 summarizes LP sizes of the eight datasets.[3]

**Synthetic datasets.** We consider three types of LPs representing packing, maximum flow, and minimum-cost flow problems, denoted by Packing, MaxFlow, and MinCostFlow, respectively. A packing problem is an LP with non-negative parameters $\boldsymbol{c}$, $\boldsymbol{A}$, and $\boldsymbol{b}$. We created a base instance by drawing their entries from the uniform distribution on $[0, 1]$ and multiplying $\boldsymbol{b}$ by $n$. We then obtained 300 random instances by multiplying all input parameters by $1 + \omega$, where $\omega$ was drawn from the uniform distribution on $[0, 0.1]$. To generate MaxFlow and MinCostFlow LPs, we first randomly created a directed graph with 50 vertices and 500 arcs and fixed source and sink vertices, denoted by $s$ and $t$, respectively. We confirmed there was an arc from $s$ to $t$ to ensure feasibility. We set base arc capacities to 1, which we perturbed by multiplying $1 + \omega$ with $\omega$ drawn from the uniform distribution on $[0, 0.1]$, thus obtaining 300 MaxFlow instances. For MinCostFlow, we set supply at $s$ and demand at $t$ to 1. We set base arc costs to 1 for all arcs but $(s, t)$, whose cost was fixed to be large enough, and perturbed them similarly using $1 + \omega$ to obtain 300 MinCostFlow instances. We transformed MaxFlow and MinCostFlow instances into equivalent inequality-form LPs with a method given in Appendix C, which requires a (trivially) feasible solution $\boldsymbol{x}_0$. For MaxFlow, we used $\boldsymbol{x}_0 = \boldsymbol{0}$ (i.e., no flow) as a trivially feasible solution. For MinCostFlow, we let $\boldsymbol{x}_0$ be all zeros but a single 1 at the entry corresponding to $(s, t)$, which is a trivially feasible (but costly) solution.

---

[2]The source code is available at https://github.com/ssakaue/data-driven-projection-lp-code.

[3]While Gurobi can solve larger LPs, we used the moderate-size LPs as the learning methods could take much longer with the limited computational resources. We admit that larger instances might introduce new challenges. Nevertheless, the trends observed in our experiments offer informative insights for larger scenarios as well.

**Realistic datasets.** We used five LPs in Netlib [15], GROW7, ISRAEL, SC205, SCAGR25, and STAIR. For each, we generated datasets of 300 random instances. To create realistic datasets, we made them contain 2% of outliers as follows. For normal 98% data points, we perturbed coefficients of objective functions by multiplying $1 + 0.1\omega$, where $\omega$ was drawn from the normal distribution; for 2% outliers, we perturbed them by multiplying $1 + \omega$, i.e., 10 times larger noises. Except for ISRAEL, the LPs have equality constraints. We transformed them into inequality-form LPs as described in Appendix C, using $x_0$ found by the initialization procedure of Gurobi's interior-point method.[4]

**Methods.** We compared four methods, named Full, ColRand, PCA, and SGA. The first two are baseline methods, while the latter two are our data-driven projection methods. Note that all four methods solved LPs with Gurobi, the state-of-the-art commercial solver. The only difference among them lies in how to reduce the dimensionality of LPs, as detailed below.

**Full:** a baseline method that returns original $n$-dimensional LPs without reducing the dimensionality.

**ColRand:** a column-randomized method based on the work by Akchen and Mišić [2], which reduces the dimensionality by selecting $k$ out of $n$ variables randomly and fixing the others to zeros.

**PCA:** the PCA-based method that reduces the dimensionality with a projection matrix $P$ learned as in Section 5.1, followed by the final projection described in Section 5.3.

**SGA:** the gradient-based method that learns $P$ as described in Section 5.2, followed by the final projection as with PCA. We initialized $P$ with that obtained by PCA and conducted a single epoch of training, setting the learning rate to $0.01$.[5]

For ColRand, PCA, and SGA, we used increasing values of the reduced dimensionality, $k = \lfloor \frac{n}{100} \rfloor, 2\lfloor \frac{n}{100} \rfloor, \ldots$, until it reached the maximum value no more than $\lfloor \frac{n}{10} \rfloor$, i.e., up to 10% of the original dimensionality. PCA and SGA learned projection matrices $P$ from $N = 200$ training instances, which were then used to reduce the dimensionality of 100 test instances. For ColRand, we tried 10 independent choices of $k$ variables and recorded the average and standard deviation.

**Results.** Figure 1 shows how the solution quality and running time of Gurobi differ among the four methods, where "objective ratio" means the objective value divided by the optimal value computed by Full. For all datasets except STAIR, PCA and/or SGA with the largest $k$ achieved about 95% to 99% objective ratios, while being about 4 to 70 times faster than Full. Regarding STAIR, PCA and SGA attained 13.1% and 51.2% objective ratios, respectively. By stark contrast, ColRand resulted in objective ratios close to zero in most cases except for Packing and ISRAEL. The results suggest that given informative training datasets, data-driven projection methods can lead to significantly better solutions than the random projection method. Regarding running times, there were differences between PCA/SGA and ColRand, which were probably caused by the numerical property of Gurobi. Nonetheless, all of the three were substantially faster than Full. In summary, the data-driven projection methods achieve high solution quality while greatly reducing the time for solving LPs.

Comparing PCA and SGA, SGA achieved better objectives than PCA in Packing, MaxFlow, Min-CostFlow, and STAIR, while performing similarly in GROW7 and SC205. In ISRAEL and SCAGR25, SGA was worse than PCA, but this is not surprising since optimizing $u(P, \pi)$ is a non-convex problem. The results suggest that no method could be universally best. Fortunately, the generalization bound (4) justifies selecting a learning method based on empirical performance. Specifically, if we adopt a learning method that produces $P$ with the best empirical performance on $N$ instances at hand, its expected performance on future instances is likely to stay within the range of $\pm\varepsilon$ of the empirical one, where $\varepsilon \lesssim H\sqrt{\mathrm{pdim}(\mathcal{U})/N} \lesssim Hk\sqrt{n/N}$ since $\mathrm{pdim}(\mathcal{U}) = \tilde{O}(nk^2)$ due to Theorem 4.4. If $N$ is sufficiently large, the high empirical performance is expected to be maintained on future instances.

Additionally, we examined the effect of the noise strength on the performance of PCA and SGA using the synthetic datasets. The details of the experiment and the results are shown in Appendix G. Therein, we found that PCA and SGA were robust against noise on capacities and costs in MaxFlow and Min-CostFlow datasets. This is probably because they can exploit fixed topologies of underlying graphs, even if the capacities and costs are largely perturbed. Fixed graph topologies are common in real-world LPs, such as those appear in transportation planning. Our data-driven projection methods can be effective in such scenarios, particularly if sufficiently large datasets of such LP instances are available.

---

[4] We expect this $x_0$ is (close to) the *analytic center*, although we could not verify Gurobi's internal processes.

[5] While we also tried SGA with the random initialization, we found that the PCA-initialization worked better. We present the results with the random initialization in Figure 3 in Appendix F for completeness.

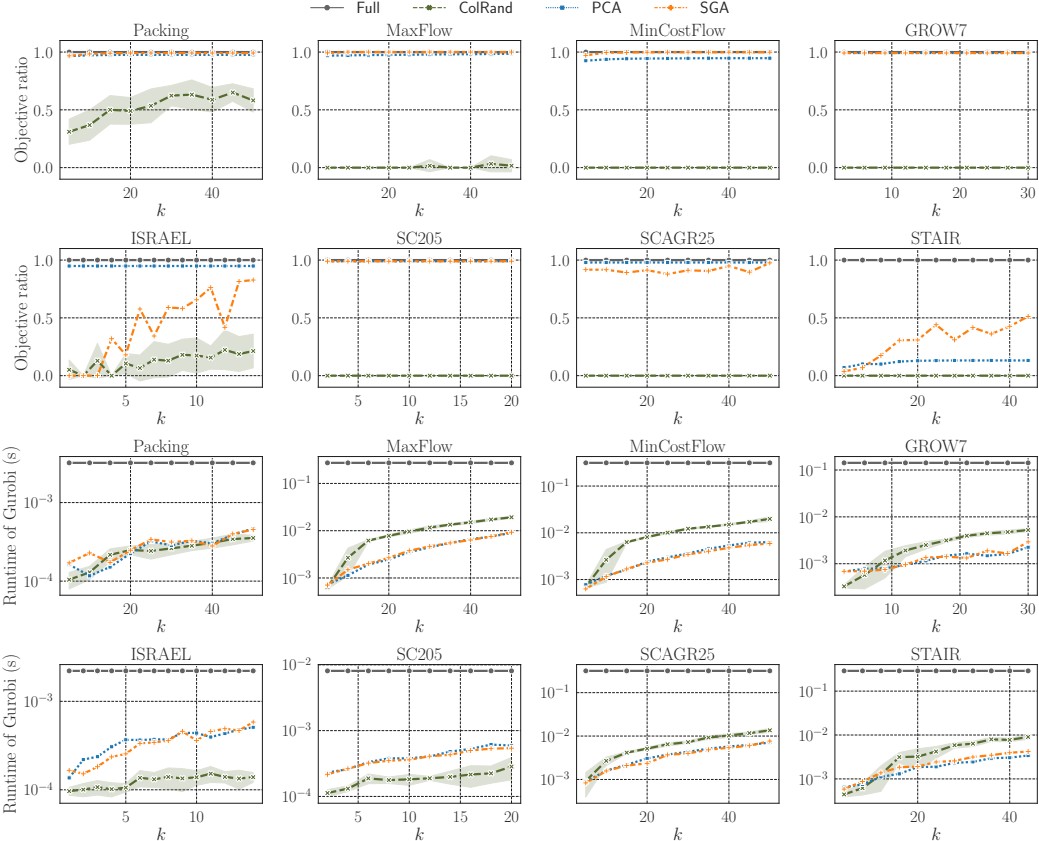

Figure 1: Plots of objective ratios (upper) and Gurobi's running times (lower, semi-log) for Full, ColRand, PCA, and SGA averaged over 100 test instances. The error band of ColRand indicates the standard deviation over 10 independent trials. The results of Full are shown for every $k$ for reference, although it always solves $n$-dimensional LPs and hence is independent of $k$.

# 7 Conclusion

We have studied the data-driven projection approach to LPs. We have established a generalization bound by proving an $\tilde{O}(nk^2)$ upper bound on the pseudo-dimension and complemented it by an $\Omega(nk)$ lower bound. We have also proposed PCA- and gradient-based learning methods and experimentally evaluated them. Our theoretical and empirical findings lay the groundwork for the further development of the data-driven approach to LPs and contribute to the broader trend of AI/ML for optimization [43].

# 8 Limitations and discussions

Our work is limited to the statistical learning setting with assumptions on LP instances (see Section 3 and Assumption 3.1). In particular, the current approach cannot deal with equality constraints varying across instances since they usually make LP instances have no common feasible solution. Despite the narrowed applicability, we believe our setting is a reasonable starting point for developing the data-driven projection approach to LPs, as is also discussed in Remark 3.2 and the paragraph following Assumption 3.1. Overcoming these limitations will require more involved methods, such as training neural networks to extract meaningful low-dimensional subspaces from non-i.i.d. messy LP instances.

Our learning methods are not efficient, and applying them to huge LPs in practice might be challenging. Similar challenges are common in most data-driven algorithm research, as discussed in Remark 5.1, and we believe our conceptually simple learning methods are helpful for future research. Regarding the data-driven approach to low-rank approximation, Indyk et al. [29] found that a few-shot learning method is useful for efficiently learning sketching matrices. A key ingredient in their method is a

surrogate loss, which enjoys a consistency guarantee and whose gradient can be computed efficiently. Empirically, they found that minimizing this loss through only a few iterations of SGD yields a good sketching matrix. We expect that similar ideas will be effective for learning projection matrices for LPs efficiently, while how to design surrogate losses in our setting is left for future work. For the same reason, our experiments are limited to moderate-size LPs, as mentioned in Footnote 3. Nevertheless, the results sufficiently serve as a proof of concept of the data-driven projection approach.

There also exist general limitations of the projection-based approach [44, 37, 2]. First, it does not consider solver-specific aspects, including numerical stability and sparsity, as discussed in Remark 2.1. Second, the projection-based approach has a limited impact on the theoretical time complexity. The theoretical time complexity of the projection-based approach is dominated by two factors: multiplying $P$ to reduce the dimensionality and solving the projected LP. Recent theoretical studies have revealed that solving an LP takes asymptotically the same computation time as matrix multiplication [17, 30], suggesting projections may not contribute to improving the total theoretical time complexity. Nevertheless, the projection-based approach leads to dramatic speedups in practice, as in Figure 1. Moreover, it can be even faster beyond the theoretical implications when GPUs are available. It is noteworthy that the projection-based approach largely benefits from GPUs, as matrix multiplication can be highly parallelized. An exciting future direction is to combine recent GPU-implemented LP solvers [6, 34, 35] with projections, which will have vast potential for solving huge LPs efficiently. Exploring data-driven projections for reducing the number of constraints will also be interesting, while this involves addressing the feasibility issue. Poirion et al. [37] used random projections for reducing the number of inequality constraints, which would provide useful insights into this direction.

### Acknowledgements

The authors thank the anonymous reviewers for their valuable comments and suggestions, particularly for inspiring us to conduct the experiments in Appendix G. This work was supported by JST ERATO Grant Number JPMJER1903, JST CREST Grant Number JPMJCR24Q2, JST FOREST Grant Number JPMJFR232L, and JSPS KAKENHI Grant Numbers JP22K17853 and 24K21315.

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

## A    Additional related work on learning through optimization

Many researchers have addressed learning tasks whose input–output pipelines involve optimization steps [4, 47, 1, 13, 42, 36, 3, 20, 45, 19]. While most of them seek to develop practical learning methods, several have studied generalization guarantees. Wang et al. [45] have studied a so-called *decision-focused learning* method with *reparametrization*, which is technically the same as projection. Their theoretical result focuses on learning of models that generate objective functions, assuming reparametrization matrices to be fixed. By contrast, we obtain a generalization bound for learning projection matrices. El Balghiti et al. [19] have studied generalization bounds in the so-called *smart predict-then-optimize* setting. While they focus on learning of models that generate coefficients of objectives from contextual information as with Wang et al. [45], our interest is in learning projection matrices, which affect both objectives and constraints. On the practical side, the line of work provides useful techniques for differentiating outcomes of optimization with respect to input parameters. Our gradient-based method for learning projection matrices is partly based on the result by Tan et al. [42].

## B    Proof of the lower bound on $\mathrm{pdim}(\mathcal{U})$

We establish the lower bound on $\mathrm{pdim}(\mathcal{U})$ in Theorem 4.5 by constructing a set of $(n-2k)k$ LP instances that $\mathcal{U}$ can shatter. The instances are written as $\pi_{r,s} = (\boldsymbol{c}_r, \boldsymbol{A}, \boldsymbol{b}_s) \in \mathbb{R}^n \times \mathbb{R}^{2k \times n} \times \mathbb{R}^{2k}$ for $r = 1, \dots, n-2k$ and $s = 1, \dots, k$, where

$$\boldsymbol{c}_r = \begin{bmatrix} \mathbf{e}_r \\ \mathbf{0}_{2k} \end{bmatrix}, \qquad \boldsymbol{A} = [\mathbf{0}_{2k,n-2k}, \mathbf{I}_{2k}], \qquad \text{and} \qquad \boldsymbol{b}_s = \begin{bmatrix} \mathbf{e}_s \\ \mathbf{0}_k \end{bmatrix}.$$

Here, $\mathbf{e}_r$ and $\mathbf{e}_s$ are the $r$th and $s$th standard basis vectors of $\mathbb{R}^{n-2k}$ and $\mathbb{R}^k$, respectively, and $\mathbf{0}_{a,b}$ is the $a \times b$ all zeros. We consider a projection matrix of the form

$$\boldsymbol{P} = \begin{bmatrix} \boldsymbol{Q} \\ \mathbf{I}_k \\ -\mathbf{I}_k \end{bmatrix},$$

where $\boldsymbol{Q} \in \{0,1\}^{(n-2k) \times k}$ is a binary matrix that we will use as tunable parameters to shatter the set of $(n-2k)k$ instances. Let $y_j$ denote the $j$th entry of the variable vector $\boldsymbol{y} \in \mathbb{R}^k$. Since we have

$$\boldsymbol{A}\boldsymbol{P} = \begin{bmatrix} \mathbf{I}_k \\ -\mathbf{I}_k \end{bmatrix}$$

the constraints, $\boldsymbol{A}\boldsymbol{P}\boldsymbol{y} \leq \boldsymbol{b}_s$, imply $y_j = 0$ for $j = 1, \dots, k$ with $j \neq s$ and $y_s \in [0,1]$. Let $\boldsymbol{y}$ be such a feasible solution. Then, the objective value is written as $\boldsymbol{c}_r^\top \boldsymbol{P}\boldsymbol{y} = \mathbf{e}_r^\top \boldsymbol{Q}\boldsymbol{y} = Q_{r,s}y_s$, where $Q_{r,s}$ is the $(r,s)$ entry of $\boldsymbol{Q} \in \{0,1\}^{(n-2k) \times k}$. Since $Q_{r,s} \in \{0,1\}$ and $y_s \in [0,1]$, we have $\max\{\boldsymbol{c}_r^\top \boldsymbol{P}\boldsymbol{y} : \boldsymbol{A}\boldsymbol{P}\boldsymbol{y} \leq \boldsymbol{b}_s\} = Q_{r,s}$. Thus, the set of those $(n-2k)k$ instances can be shattered by setting all threshold values to $1/2$ and appropriately choosing each entry of $\boldsymbol{Q} \in \{0,1\}^{(n-2k) \times k}$. In other words, all the $2^{(n-2k)k}$ outcomes of $\{u(\boldsymbol{P}, \pi_{r,s}) = Q_{r,s} \geq 1/2 : r = 1, \dots, n-2k, \ s = 1, \dots, k\}$ can realize by changing $\boldsymbol{P} \in \mathbb{R}^{n \times k}$ (or $\boldsymbol{Q} \in \{0,1\}^{(n-2k) \times k}$). Thus, we obtain an $\Omega(nk)$ lower bound on the pseudo-dimension of $\mathcal{U} = \{u(\boldsymbol{P}, \cdot) : \Pi \to \mathbb{R} : \boldsymbol{P} \in \mathbb{R}^{n \times k}\}$.

## C    How to remove equality constraints

Suppose that we are given an LP of the form

$$\underset{\boldsymbol{z} \in \mathbb{R}^n}{\text{maximize}} \quad \boldsymbol{w}^\top \boldsymbol{z} \qquad \text{subject to} \quad \boldsymbol{A}_{\text{ineq}}\boldsymbol{z} \leq \boldsymbol{b}_{\text{ineq}}, \ \boldsymbol{A}_{\text{eq}}\boldsymbol{z} = \boldsymbol{b}_{\text{eq}},$$

which has both inequality and equality constraints. Below, assuming that a (trivially) feasible solution $\boldsymbol{x}_0$ is available (i.e., $\boldsymbol{A}_{\text{ineq}}\boldsymbol{x}_0 \leq \boldsymbol{b}_{\text{ineq}}$ and $\boldsymbol{A}_{\text{eq}}\boldsymbol{x}_0 = \boldsymbol{b}_{\text{eq}}$), we transform the LP into an equivalent inequality form. First, we replace the variable vector $\boldsymbol{z}$ with $\boldsymbol{z}' + \boldsymbol{x}_0$, obtaining an equivalent LP of the form

$$\underset{\boldsymbol{z}' \in \mathbb{R}^n}{\text{maximize}} \quad \boldsymbol{w}^\top (\boldsymbol{z}' + \boldsymbol{x}_0) \qquad \text{subject to} \quad \boldsymbol{A}_{\text{ineq}}\boldsymbol{z}' \leq \boldsymbol{b}_{\text{ineq}} - \boldsymbol{A}_{\text{ineq}}\boldsymbol{x}_0, \ \boldsymbol{A}_{\text{eq}}\boldsymbol{z}' = \mathbf{0}.$$

The equality constraints, $A_{\mathrm{eq}} z' = 0$, mean that $z'$ must be in the null space of $A_{\mathrm{eq}}$. Therefore, $z'$ is always represented as $z' = (I - A_{\mathrm{eq}}^\dagger A_{\mathrm{eq}}) x$ with some $x \in \mathbb{R}^n$, where $A_{\mathrm{eq}}^\dagger$ is the pseudo-inverse of $A_{\mathrm{eq}}$ and $I - A_{\mathrm{eq}}^\dagger A_{\mathrm{eq}}$ is the orthogonal projection matrix onto the null space of $A_{\mathrm{eq}}$.[6] Substituting $z' = (I - A_{\mathrm{eq}}^\dagger A_{\mathrm{eq}}) x$ into the above LP allows us to remove the equality constraints since they are automatically satisfied for every $x$. After all, by transforming the variable vector as $z = (I - A_{\mathrm{eq}}^\dagger A_{\mathrm{eq}}) x + x_0$, we can obtain an equivalent LP of the form

$$\underset{x \in \mathbb{R}^n}{\text{maximize}} \quad w^\top (I - A_{\mathrm{eq}}^\dagger A_{\mathrm{eq}}) x \qquad \text{subject to} \quad A_{\mathrm{ineq}} (I - A_{\mathrm{eq}}^\dagger A_{\mathrm{eq}}) x \leq b_{\mathrm{ineq}} - A_{\mathrm{ineq}} x_0,$$

where the additive constant, $w^\top x_0$, in the objective is omitted. This is an inequality-form LP (1) with $c = (I - A_{\mathrm{eq}}^\dagger A_{\mathrm{eq}})^\top w$, $A = A_{\mathrm{ineq}} (I - A_{\mathrm{eq}}^\dagger A_{\mathrm{eq}})$, and $b = b_{\mathrm{ineq}} - A_{\mathrm{ineq}} x_0$. Note that $x = 0$ is always feasible for the resulting LP and that we can use this transformation if $A_{\mathrm{eq}}$ and $b_{\mathrm{eq}}$ are fixed, even if $w$, $A_{\mathrm{ineq}}$, and $b_{\mathrm{ineq}}$ can change across instances.

## D  Derivation of the gradient in SGA

We explain how to derive the gradient of $u(P, \pi)$ in (5), which indeed follows from Tan et al. [42, Theorem 1] (or the implicit function theorem). To focus on computing the gradient, we suppose that the columns of $P$ have already been projected onto the feasible region of $\pi$ by the projection step (6).

Consider computing the gradient $\nabla u(P, \pi)$ of $u(P, \pi) = \max\{ c^\top P y : A P y \leq b \}$ with respect to $P$. For convenience, we define new parameters $w = P^\top c \in \mathbb{R}^k$ and $W = A P \in \mathbb{R}^{m \times k}$ and let $y^* \in \arg\max\{ w^\top y : W y \leq b \}$. Then, we can differentiate the optimal value, $u(P, \pi) = w^\top y^*$, with respect to $w$ and $W$ by applying the implicit function theorem to the KKT condition. Specifically, as shown in Tan et al. [42, Theorem 1], we have $\frac{\partial u}{\partial w} = y^*$ and $\frac{\partial u}{\partial W} = -\lambda^* y^{*\top}$, where $\lambda^* \in \mathbb{R}_{\geq 0}^m$ is the dual optimal solution. From the chain rule, we have

$$\nabla u(P, \pi) = \frac{\partial u}{\partial w} \cdot \frac{\partial w}{\partial P} + \frac{\partial u}{\partial W} \cdot \frac{\partial W}{\partial P},$$

where the indices for the products are aligned appropriately. By substituting the derivatives into this, we obtain $\nabla u(P, \pi) = c y^{*\top} - A^\top \lambda^* y^{*\top}$. When applying the implicit function theorem, we must ensure that the Jacobian matrix is invertible. In the above case, ensuring the regularity condition (i.e., active constraints are linearly independent at $y^*$) is sufficient.

## E  Running time of learning methods

We discuss the theoretical complexity of the PCA- and SGA-based methods. For convenience, we use $T_{\mathrm{lp}}(m, n)$ to represent the time complexity of solving an LP instance with $m$ inequality constraints and $n$ variables, as this factor highly depends on problem settings. Also, let $T_{\mathrm{proj}}(m)$ be the time for solving the problem in (6) $k$ times for projecting columns of $P \in \mathbb{R}^{n \times k}$ onto the feasible region specified by $m$ inequality constraints. Given the $T_{\mathrm{lp}}(m, n)$-time linear optimization oracle, we can implement this projection step with a Frank–Wolfe-style algorithm. In this case, the time for projecting columns of $P$ within an $\varepsilon$-error is typically $T_{\mathrm{proj}}(m) = T_{\mathrm{lp}}(m, n) \cdot \mathrm{poly}(n, m) \log(1/\varepsilon)$ [32, 22].

**PCA-based method.** Computing SVD of $X \in \mathbb{R}^{N \times n}$ takes $\mathrm{O}(N n^2)$ time. Then, the final projection takes up to $T_{\mathrm{proj}}(Nm)$ time. Thus, the total time complexity is $\mathrm{O}(N n^2) + T_{\mathrm{proj}}(Nm)$.

**SGA-based method.** We discuss the complexity of a single iteration, which consists of projecting columns of $P$ as in (6), solving a projected LP for obtaining $y^*$ and $\lambda^*$, and computing the gradient in (5). These take $T_{\mathrm{proj}}(m)$, $T_{\mathrm{lp}}(m, k)$, and $\mathrm{O}(n(m + k))$ time, respectively. Thus, the per-iteration complexity is $T_{\mathrm{proj}}(m) + T_{\mathrm{lp}}(m, k) + \mathrm{O}(n(m + k))$. After finishing all the iterations, the final projection takes $T_{\mathrm{proj}}(Nm)$ time, as with the PCA-based method. In the experiments, we ran SGA for a single epoch, i.e., $N$ iterations. Thus, the total time complexity is $N(T_{\mathrm{proj}}(m) + T_{\mathrm{lp}}(m, k) + \mathrm{O}(n(m + k))) + T_{\mathrm{proj}}(Nm)$.

---

[6]We can also represent $z'$ by a linear combination of a basis of the null space of $A_{\mathrm{eq}}$, which we can compute via SVD. However, we found that this representation was numerically unstable in our experiments.

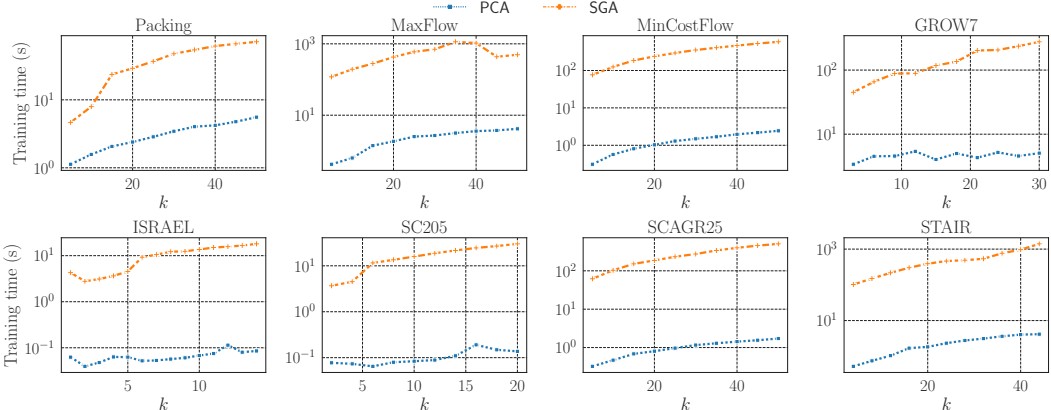

Figure 2: Running times of PCA and SGA for learning projection matrices on 200 training instances.

We turn to the running times of the PCA- and SGA-based methods in the experiments in Section 6. Figure 2 shows the times taken by PCA and SGA for learning projection matrices on training datasets of 200 instances. (Full and ColRand are not included since they do not learn projection matrices.) We assumed that optimal solutions of training instances were computed a priori, and hence the time for solving original LPs was not included. The figure shows that SGA took much longer than PCA. This is natural since SGA iteratively solves LPs for computing gradients (5) and quadratic programs for projection (6), while PCA only requires computing the top-$(k-1)$ right-singular vectors of $\boldsymbol{X} - \boldsymbol{1}_N \bar{\boldsymbol{x}}^\top$, as discussed above.

## F  Results with random initialization of SGA

Figure 3 shows the same plots as in Figure 1 but with SGA initialized with ColRand instead of PCA, which was mentioned in Footnote 5.

## G  Objective ratios on synthetic datasets with various noise levels

This section examines the effect of the noise strength on the performance of our data-driven projection methods. We created synthetic datasets, Packing, MaxFlow, and MinCostFlow, in the same way as in Section 6. An important difference from Section 6 is the increased noise level $\omega$, which perturbs LP inputs through multiplication by $1 + \omega$. We draw $\omega$ from a uniform distribution over $[0, \bar{\omega}]$ with the upper bound $\bar{\omega}$ ranging from 0.0 to 2.0 in increments of 0.2; in Section 6, $\bar{\omega}$ was fixed at 0.1. The larger $\bar{\omega}$ is, the less consistent the tendencies in the LP input parameters become, making it more challenging to learn projection matrices with PCA and SGA. We fixed the dimensionality $k$ of projected LPs to 20, which means the size of projection matrices $\boldsymbol{P}$ is $n \times 20$.

Figure 4 presents objective ratios achieved by each method on Packing, MaxFlow, and MinCostFlow datasets. There is a notable difference between Packing and the others, which we discuss below.

**Packing.** The performance of our data-driven methods (PCA and SGA) worsens as $\bar{\omega}$ increases, as expected. While they exhibit clear advantage over the random-projection baseline (ColRand) at small $\bar{\omega}$ values, they behave similarly to ColRand when $\bar{\omega} = 2.0$.

**MaxFlow and MinCostFlow.** In contrast to the Packing case, our data-driven methods, particularly SGA, performed well even with high noise levels, while the performance of ColRand remained poor. The success of data-driven methods is probably due to the fixed graph topology, which creates consistent tendencies across LP instances despite varying edge capacities and costs.

Note that fixed graph topologies are ubiquitous. For example, in daily transportation planning, the topology of traffic networks is fixed, while the capacities and costs may fluctuate due to congestion. The above results highlight the merit of our data-driven projection methods in such applications.

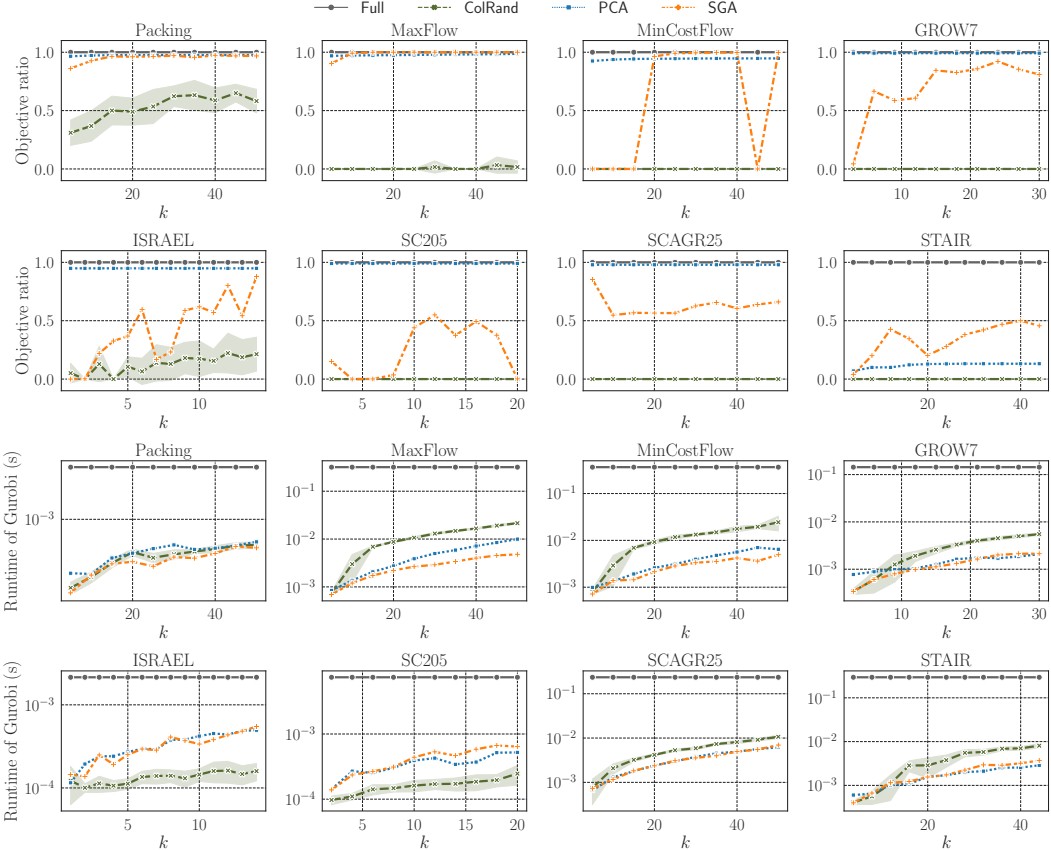

Figure 3: The same plots as in Figure 1 but with SGA initialized with ColRand instead of PCA, as mentioned in Footnote 5. Compared with Figure 1, the objective ratio of SGA deteriorates particularly in MinCostFlow, GROW7, SC205, and SCAGR25.

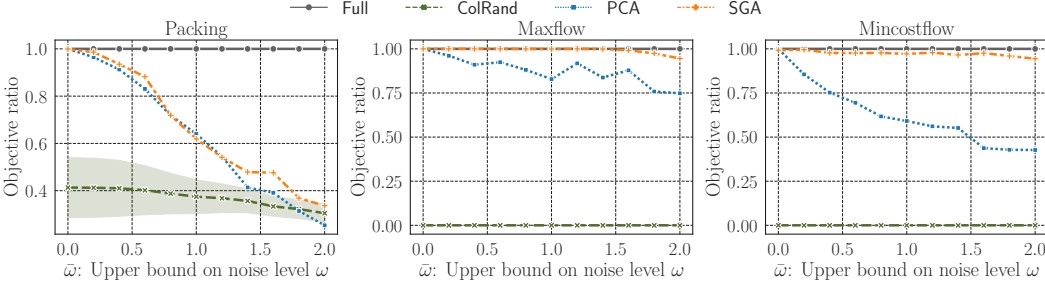

Figure 4: Objective ratios on synthetic datasets with varying upper bounds $\bar{\omega}$ on the noise level. Except for Full, the LP dimensionality is set to $k = 20$. Other settings are the same as in Section 6.

