# OpenReview forum: "Generalization Bound and Learning Methods for Data-Driven Projections in Linear Programming"
_NeurIPS.cc/2024/Conference — NeurIPS 2024 poster_

### Official Review · Reviewer_hECG · 2024-07-06

**Soundness:** 3
**Presentation:** 3
**Contribution:** 2
**Rating:** 6
**Confidence:** 4

**Summary:**

On the theoretical side, the paper studies the problem of sample complexity of learning data-driven projection matrices for accelerating high-dimensional LP solving. Given n-dimensional LPs drawn from some problem distribution, the goal is to bound the number of problem instances needed to learn an $n\times k$ projection matrix such that a good solution to the original LP can be found by (efficiently) solving the k-dimensional projected LP. The main result is a $\tilde{O}(nk^2)$ upper bound on pseudo-dimension of the class of functions that measure the optimal value of the projected LP, and a corresponding $\Omega(nk)$ lower bound.

While finding optimal projection matrix on a training set is hard, the paper proposes two practical approaches for learning a projection matrix from the training LP instances. The first approach applies Principle Components Analysis to a matrix of optimal solutions of training instances. The second approach uses stochastic gradient updates to learn the projection matrix. Empirically, data-driven projections seem to perform better than random projections, with comparable gains in speed-ups.

**Strengths:**

- Speeding up high-dimensional LPs is fundamental to operations research and data-driven projections appears to be a promising tool.
- A polynomial upper bound on the sample complexity theoretically establishes that projection matrices are learnable from data.
- There is also a lower bound, with is tight up to a factor of $k$ and logarithmic factors.
- Even though exact optimization on training LP instances is hard, the paper provides practical solutions that work well empirically. Practical algorithms are often hard to design in the data-driven literature.

**Weaknesses:**

- The i.i.d. assumption needed in theoretical results may be too strong in practice.
- The proposed methods for learning the projection matrix are not efficient e.g. PCA approach needs optimal solutions of training problem instances.
- While the authors provide bounds on the time complexity of their learning methods, there are no guarantees about the quality of solutions.

**Questions:**

- How does the proof of Lemma 4.3 differ from the techniques of Balcan et al., NeurIPS 2022?
- It seems like the proof for upper bound can be simplified by using the GJ framework of Bartlett et al. COLT 2022 (P. L. Bartlett, P. Indyk, and T. Wagner. Generalization bounds for data-driven numerical linear algebra.)
- The $\log (H/\varepsilon)$ term is not needed in the sample complexity bound $N$. (Line 153)
- Line 209: sign patters

**Limitations:**

Yes.

---

> ### Author Rebuttal · Authors · 2024-08-04
>
> We sincerely thank the reviewer for providing invaluable comments based on a deep understanding of data-driven algorithm design. We respond to each comment below:
>
> >Weaknesses:
> >- The i.i.d. assumption needed in theoretical results may be too strong in practice.
> >- The proposed methods for learning the projection matrix are not efficient e.g. PCA approach needs optimal solutions of training problem instances.
> >- While the authors provide bounds on the time complexity of their learning methods, there are no guarantees about the quality of solutions.
>
> We acknowledge these points as limitations of our work. However, we wish to highlight that these significant challenges are not unique to our work but are common in data-driven algorithm design. Most existing research in this area assumes the i.i.d. setting and often does not offer efficient learning methods nor assess the quality of the resulting outputs. There are a few notable exceptions, such as the work by Balcan et al. (2023), titled "Output-sensitive ERM-based techniques for data-driven algorithm design." Such enumeration-based methods can learn parameters that minimize empirical risk but lack polynomial-time guarantees and may be impractical due to their substantial computational demands. In contrast, our methods for learning projection matrices, although not providing guarantees regarding solution quality, have been experimentally demonstrated to perform well. We believe that providing such methods contributes to further investigation into the practical aspect of data-driven algorithm design.
>
> > Questions:
> > - How does the proof of Lemma 4.3 differ from the techniques of Balcan et al., NeurIPS 2022?
>
> We appreciate this insightful question. While the work of Balcan et al. (NeurIPS 2022), which we cited as [11] in our paper, indeed inspired our analysis, there is an important technical difference between their approach and ours. Balcan et al. focus specifically on the branch-and-cut method, analyzing situations where either (1) new cuts (constraints) do not separate an optimal solution $x^*_{\rm LP}$ to the original LP or (2) new cuts separate $x^*_{\rm LP}$ and constitute an equation system specifying a new optimal solution.
>
> In contrast, our Lemma 4.3 is intended for investigating the behavior of optimal values of projected LPs $(\boldsymbol{P^\top c}, \boldsymbol{AP}, \boldsymbol{b})$, which can change entirely with projection matrix $\boldsymbol{P}$. A particular challenge we address is the possibility of $\boldsymbol{AP}$ becoming rank-deficient, which hinders the standard derivation of an equation system specifying an optimal solution to the projected LP (cf. the proof of Korte and Vygen [31, Proposition 3.1]). We circumvent this issue by reformulating the projected LP into an equivalent $2k$-dimensional LP with non-negativity constraints, as is done at the beginning of the proof of Lemma 4.3. This adjustment, not employed by Balcan et al., justifies the subsequent analysis based on the enumeration of vertex solutions in the feasible region of the projected LP. Although the adjustment is common in mathematical programming, using it for analyzing the pseudo-dimension would be a novel idea.
>
> Thus, although both our analysis and that of Balcan et al. employ Cramer’s rule and might seem similar at first glance, there is a notable technical distinction in how we address the specific challenges in analyzing projected LPs.
>
> > - It seems like the proof for upper bound can be simplified by using the GJ framework of Bartlett et al. COLT 2022 (P. L. Bartlett, P. Indyk, and T. Wagner. Generalization bounds for data-driven numerical linear algebra.)
>
> We greatly value this insightful comment. Indeed, we considered employing the GJ framework of Bartlett et al. (COLT 2022) as an alternative approach, but we encountered a difficulty. The GJ framework requires a *GJ algorithm* that, in our case, computes the optimal value $u(\boldsymbol{P}, \pi)$ up to an $\varepsilon$-error for a sufficiently small $\varepsilon>0$. Crucially, the *predicate complexity* of a GJ algorithm must be upper bounded independently of projection matrix $\boldsymbol{P}$ (i.e., learnable parameters). If not, certain parameters $\boldsymbol{P}$ may cause the predicate complexity to become arbitrarily large, preventing us from bounding the pseudo-dimension with the GJ framework. We attempted to develop such a GJ algorithm using simplex- and interior-point-type methods, yet we were unable to make their predicate complexity independent of $\boldsymbol{P}$.
>
> We also appreciate the comments about the $\log(H/\varepsilon)$ term and the typo.
>
> We hope our responses above have adequately addressed the reviewer's concerns and questions. Please do not hesitate to contact us during the discussion period if there are any further questions.

---

> > ### Comment · Reviewer_hECG · 2024-08-09
> >
> > Thank you for the detailed response, I appreciate the answers to my questions. I retain my score.

---

### Official Review · Reviewer_E5r3 · 2024-07-09

**Soundness:** 3
**Presentation:** 3
**Contribution:** 2
**Rating:** 6
**Confidence:** 3

**Summary:**

This paper considers data-driven approach for learning projections for LPs. Given an LP with $m$ constraints and $n$ variables, it establishes bound on learning a projection $P\in \mathbb{R}^{n\times k}$ that reduces $n$ to $k$. The main contribution is to establish uniform convergence bound on pseudo-dimension, for an upper bound of $\tilde O(nk^2)$ and a lower bound of $\Omega(nk)$. Authors also propose two algorithms for learning the projection, one is based on PCA of the optimal solution of LP instances, the other is a gradient ascent approach.

**Strengths:**

This paper establishes bound on pseudo-dimension of of performance metrics for data-driven projections for LP, the bound is tight up to a factor of $k$. Experiments are also performed for the two methods proposed in this paper, and it provides speed up over the column-randomized approach that is data-oblivious and performance upgrade.

**Weaknesses:**

The techniques for proving the upper and lower bounds on pseudo-dimension are quite standard, authors should emphasize the technical difficulties for proving these bounds.

Empirical side, while "after* learning the projection, it gives good performance, the process of learning the projection is quite slow. Authors might consider adding more discussions on how to learn these projections more efficiently (from an algorithmic perspective).

**Questions:**

Suppose you learn the projection for both the primal and dual, could this lead to an even more efficient downstream algorithm, as one only needs to handle LP instance of size $k'\times k$? Or are there obvious reasons this is not a good idea?

**Limitations:**

Yes.

---

> ### Author Rebuttal · Authors · 2024-08-04
>
> We sincerely thank the reviewer for providing insightful comments and a positive evaluation. We respond to each comment below.
>
> > Weaknesses:
> >
> > The techniques for proving the upper and lower bounds on pseudo-dimension are quite standard, authors should emphasize the technical difficulties for proving these bounds.
>
> We appreciate the reviewer's suggestion. A primary technical challenge lies in the proof of Lemma 4.3, which is pivotal to our solver-agnostic analysis of optimal values of LPs. Specifically, we investigate the behavior of the optimal value of the projected LP $(\boldsymbol{P^\top c}, \boldsymbol{AP}, \boldsymbol{b})$ while addressing the potential issue that $\boldsymbol{AP}$ may be rank-deficient. This rank deficiency generally hinders the straightforward derivation of an equation system that specifies a vertex optimal solution. To overcome this, the proof of Lemma 4.3 begins by reformulating the LP into an equivalent $2k$-dimensional LP with non-negativity constraints. This adjustment facilitates the determination of equation systems specifying vertex optimal solutions, as in the proof of Korte and Vygen [31, Proposition 3.1]. Although the adjustment is common in the context of mathematical programming, using it for analyzing the pseudo-dimension is our novel idea. We will emphasize this technical nuance in the revised manuscript.
>
> > Empirical side, while "after* learning the projection, it gives good performance, the process of learning the projection is quite slow. Authors might consider adding more discussions on how to learn these projections more efficiently (from an algorithmic perspective).
>
> We greatly value this suggestion. We expect that the few-shot learning approach, akin to the method proposed by Indyk et al. (NeurIPS 2021) for data-driven low-rank approximation, is effective for learning projection matrices efficiently. We will expand this discussion, currently only briefly mentioned in the conclusion section, in our revised manuscript.
>
> > Questions:
> >
> > Suppose you learn the projection for both the primal and dual, could this lead to an even more efficient downstream algorithm, as one only needs to handle LP instance of size $k' \times k$? Or are there obvious reasons this is not a good idea?
>
> We appreciate this insightful question. Indeed, we have considered applying projections to both the primal and dual and recognize its potential benefits. However, we have opted not to reduce the dual variables (i.e., the number of constraints), as it could result in optimal solutions for projected LPs that are infeasible for the original LPs. While the quality of feasible solutions is naturally evaluated through their objective values, how to assess the quality of infeasible solutions is more controversial. For this reason, we have chosen to focus solely on reducing the number of primal variables, thereby avoiding the issue of infeasibility and maintaining conceptual simplicity. We value this discussion and will highlight it as a promising direction for future research.
>
> We hope our responses have adequately addressed the reviewer's concerns and questions. Please do not hesitate to contact us during the discussion period if there are any further questions.

---

> > ### Comment · Reviewer_E5r3 · 2024-08-07
> >
> > I thank the authors for answering my questions. I'll keep my score as is.

---

### Official Review · Reviewer_nj7Z · 2024-07-10

**Soundness:** 3
**Presentation:** 3
**Contribution:** 2
**Rating:** 5
**Confidence:** 3

**Summary:**

The paper attempts to theoretically analyze a method called Data-Driven Projections in Linear Programming. As discussed in the paper, projection methods aim to reduce the size of high-dimensional LPs. While random projection methods have improved the efficiency of LPs, data-driven projections have achieved better results. In this paper, it is assumed that the parameters of the LP (parameters in the constraints and the objective) come from some distribution. Based on a set of training samples of these parameters, data-driven methods try to find a subspace where future optimal solutions are expected to appear. The main contribution of the paper is that they propose a generalization bound for the difference between the empirical and statistical performance of the projected LP.

**Strengths:**

- The paper is well-written and well-motivated.
- The proofs are rigorous and well-written.

**Weaknesses:**

- I think the contribution of the paper is marginal.
- Although it is mentioned in the paper that the generalization is independent of the choice of the projection matrix $P$, the final goal is to minimize the expected value of the objective. To achieve this, it is necessary to choose a good $P$ that minimizes the empirical optimal value. I believe it should be proven somehow that the algorithm proposed by the paper can suggest a projection matrix that captures the subspace where future optimal solutions are expected to appear. I couldn't find a proof for this in the paper.

**Questions:**

Please refer to the weaknesses section.

**Limitations:**

Please refer to the weaknesses section.

---

> ### Author Rebuttal · Authors · 2024-08-04
>
> We are grateful for the reviewer's valuable feedback. We present our response to the comments below.
>
> > Weaknesses:
> > - I think the contribution of the paper is marginal.
> > - Although it is mentioned in the paper that the generalization is independent of the choice of the projection matrix $P$, the final goal is to minimize the expected value of the objective. To achieve this, it is necessary to choose a good $P$ that minimizes the empirical optimal value. I believe it should be proven somehow that the algorithm proposed by the paper can suggest a projection matrix that captures the subspace where future optimal solutions are expected to appear. I couldn't find a proof for this in the paper.
>
> We appreciate the reviewer's insights provided. Regarding the second point, finding $\boldsymbol{P}$ that minimizes the empirical optimal value (i.e., empirical risk minimization, or ERM) is indeed our ultimate goal. We attempted to prove such guarantees, but it turned out challenging as the optimal value of a projected LP, viewed as a function of $\boldsymbol{P}$, is non-convex and difficult to optimize directly. Please note that this is a general difficulty recognized in the field of *data-driven algorithm design*. Accordingly, the line of work in this area (e.g., Gupta & Roughgarden SICOMP2017; Balcan et al. STOC2021; Bartlett et al. COLT2022), including ours, aims to achieve generalization bounds that are independent of tunable parameters, i.e., *uniform convergence*. This shift from seeking optimality in ERM to embracing uniform convergence is a deliberate strategy reached after careful consideration in this research field. Therefore, we believe that this point should not undermine the extensive body of work, including ours.
>
> Just to make sure, we wish to re-emphasize the discussion in Remark 4.2: uniform convergence ensures that $\boldsymbol{P}$ performing well on training instances is also expected to perform well on future instances, *even if $\boldsymbol{P}$ does not minimize the empirical optimal value.* Furthermore, the experiments in Section 6 show that we can empirically find good $\boldsymbol{P}$ with PCA- and SGA-based methods, whose generalization guarantee follows from uniform convergence. These theoretical and empirical results demonstrate the merit of our data-driven projection approach to LPs, effectively circumventing the aforementioned challenge of optimizing $\boldsymbol{P}$ to minimize the empirical optimal value.
>
> We hope that the above clarifications have adequately addressed the reviewer's concerns.
>
> > Questions:
> >
> > Please refer to the weaknesses section.
>
> In response to the first comment in the weaknesses section, "I think the contribution of the paper is marginal," we would appreciate it if the reviewer could provide more details about this opinion during the discussion period, particularly if our clarifications above have not fully addressed the reviewer's concern.

---

> > ### Comment · Reviewer_nj7Z · 2024-08-10
> >
> > I thank the authors for their answers. They addressed my questions, so I increased my score accordingly.

---

### Official Review · Reviewer_vPup · 2024-07-16

**Soundness:** 3
**Presentation:** 3
**Contribution:** 3
**Rating:** 6
**Confidence:** 4

**Summary:**

The paper proposes a data-driven approach to an accelerated solution of linear programming problems belonging to a common family. To this end, the dimensionality of the problems is reduced by a projection learned from a training set of problems. The paper first gives a theoretical generalization bound for this learning problem. Also, the paper proposes two specific projection learning algorithms, based on PCA and stochastic gradient ascent. Finally, the performance of the proposed methods is experimentally illustrated on several test cases.

**Strengths:**

The paper is generally quite nicely written and very readable. I could follow it everywhere without difficulty and didn't even notice any typos.

The theoretical generalization bound (Theorem 4.4) is derived using more or less standard approaches of statistical learning theory, but the result itself is new and the proof seems to be correct. The proof is presented in the main text and is concise and nice.

The idea of data-driven projection and its implementations based on PCA and gradient ascent seem to be original. The paper includes a comparison of the proposed algorithm on 8 test cases; the proposed algorithms show good performance there.

**Weaknesses:**

The most significant issue that I see is that the derived generalization bound is vacuous for the LP families experimentally studied in the paper, but this point is for some reason completely ignored by the authors. Specifically, the bound in question is $\epsilon\lesssim Hk\sqrt{n/N}$ (lines 257 and 359). As far as I understand, in the experiments $N\sim 200, n\sim 500, k\sim 20$. Substituting these numbers gives $\epsilon\lesssim 20H,$ which is vacuous because the expected performance of a solver is in the interval $[0,H]$ anyway. Not to mention that the $O(\cdots)$ in the bound may contain additional large constants. In fact, asymptotic generalization bounds are well-known to (significantly) overestimate the true generalization gap and so are more suitable as a mere theoretical assurance of convergence. However, the paper seems to suggest (say, in the end of section 6) that the derived bound has some practical value while making no attempt to actually discuss specific numbers associated with its experiments, which I find completely misleading. (This does not undermine the theoretical merit of Theorem 4.4; on the other hand convergence rates $O(\sqrt{pdim/N})$ are standard in SLT.)

The second general significant issue is that the whole setting of data-learnable projected optimization proposed in the paper seems fairly artificial to me. This setting obviously assumes the variables and constraints to be somehow aligned between different LP instances (in contrast, random projections do not require any such alignment). This point is discussed in Remark 3.2, where it is mentioned that such scenarios arise in daily production planning and flight scheduling. It would be interesting to see a particular real scenario of this type. All the example experimentally studied in the paper, even those that the authors call realistic, do not look realistic to me. As far as I understood, each of the 8 considered families is obtained by synthetically perturbing a single LP instance. This artificial setting is obviously very favorable to the proposed algorithms compared to the baselines. If the perturbation goes to 0, the family degenerates into a collection of identical LP instances that can all be perfectly "solved" by recalling a single solution, so the proposed training-based methods can be made to look arbitrarily more efficient than the baselines by tuning the perturbation magnitude.

**Questions:**

N/A

---

> ### Author Rebuttal · Authors · 2024-08-04
>
> We are truly grateful for the reviewer's thoughtful and inspiring feedback. Below we present our responses to the comments.
>
> > Weaknesses:
> >
> > The most significant issue that I see is that the derived generalization bound is vacuous for the LP families experimentally studied in the paper, [...] However, the paper seems to suggest (say, in the end of section 6) that the derived bound has some practical value while making no attempt to actually discuss specific numbers associated with its experiments, which I find completely misleading.
>
> We apologize for any misleading expressions, particularly the sentence at the end of Section 6. The primary purpose of the experiments was to observe the empirical performance of projection matrices learned with the PCA- and SGA-based methods, rather than to evaluate the sharpness of the bound in Theorem 4.4. We intended to communicate that the bound of $\varepsilon \lesssim Hk\sqrt{n/N}$ on the generalization error could be meaningful when $N$ is sufficiently large. Although using such a large training dataset may seem demanding, it is a plausible future scenario given the trend towards accumulating and utilizing more data to advance optimization technologies, as evidenced by projects like AI4OPT. We will revise the manuscript to clarify this point and avoid any misconceptions. We appreciate the reviewer pointing out this issue.
>
> To merely supplement our revised statements, we conducted an additional experiment, shown in Additional Experiment 2 in the [global response](https://openreview.net/forum?id=jHh804fZ5l&noteId=wB14h9cLk0), using a larger dataset of smaller LP instances for the Packing problem. The dataset consists of $20,000$ training and $20,000$ test LP instances of size $(n, m) = (50, 5)$, and we set the reduced dimensionality $k$ to $2$. Substituting these parameters into the bound implies $\varepsilon \lesssim Hk\sqrt{n/N} = 0.1H$, which is now non-vacuous. We computed approximate generalization errors (the left-hand side in Eq. 4) for projection matrices learned with PCA- and SGA-based methods, where the true expectation is approximated by taking an average over the $20,000$ test instances. Figure 2 in the global response compares these to the reference bound of $k\sqrt{n/N}$ implied by the theoretical analysis (omitting log and constant factors). While empirical generalization errors are typically much better than what the theory implies, all curves show a decreasing trend, and the theoretical upper bound converges to zero as $N$ increases, which could offer meaningful bounds when $N$ is large. We hope this clarification effectively addresses the reviewer's concern.
>
> > (This does not undermine the theoretical merit of Theorem 4.4; on the other hand convergence rates $O(\sqrt{pdim/N})$ are standard in SLT.)
>
> Just to clarify, we wish to emphasize that the fact that $O(\sqrt{{\rm pdim(\mathcal{U})}/N})$ is standard in statistical learning theory also does not undermine the contribution of Theorem 4.4. The contribution of Theorem 4.4 lies in establishing the bound on the pseudo-dimension as ${\rm pdim(\mathcal{U})}=O(nk^2\log mk)$, not in asserting the $O(\sqrt{{\rm pdim(\mathcal{U})}/N})$ bound on the generalization error. The latter is used merely as a standard fact in our paper.
>
> > The second general significant issue is that the whole setting of data-learnable projected optimization proposed in the paper seems fairly artificial to me. [...] This artificial setting is obviously very favorable to the proposed algorithms compared to the baselines. If the perturbation goes to 0, the family degenerates into a collection of identical LP instances that can all be perfectly "solved" by recalling a single solution, so the proposed training-based methods can be made to look arbitrarily more efficient than the baselines by tuning the perturbation magnitude.
>
> We acknowledge the reviewer's points and appreciate the insights provided. While attempting to obtain more realistic datasets, we found that publicly available repositories such as Netlib, which we used in our experiments, only offer a single LP instance for each setting and do not provide the volume of data necessary for training. Thus, we created datasets by adding noise. Please note that our realistic datasets include outliers, making them less artificial than the reviewer might have perceived.
>
> To address the reviewer's concern that our datasets might be too favorable to our proposed methods, we conducted additional experiments with the Packing, Maxflow, and MinCostFlow datasets at higher noise levels; please refer to Additional Experiment 1 in the [global response](https://openreview.net/forum?id=jHh804fZ5l&noteId=wB14h9cLk0) for details. Since data-driven methods can derive no benefit from completely noisy data, increasing the noise level effectively creates more challenging datasets.
>
> Figure 1 in the PDF attached to the global response presents the results. For Packing LPs, higher noise levels led to poorer performance of data-driven methods (PCA and SGA), aligning with the reviewer's perspective regarding our methods applied to less favorable settings. By contrast, for MaxFlow and MinCostFlow LPs, data-driven methods, particularly SGA, continued to perform well even in highly noisy environments. As described in the global response, this robustness is probably attributed to the fixed graph topology. Since fixed graph topologies are prevalent in practical scenarios such as transportation planning on traffic networks, we believe the noise resistance in these datasets indicates substantial additional merit of our methods.
>
> This unexpected finding underscores the value of the reviewer's feedback, which has helped us present another positive aspect of our approach. We hope our responses adequately address the reviewer's concerns and present our work in a more positive light. Please do not hesitate to reach out during the discussion period if there are any further questions.

---

> > ### Comment · Reviewer_vPup · 2024-08-10
> >
> > I thank the authors for their reply and addressing both of my concerns. I'm raising my score.

---

### Author Rebuttal · Authors · 2024-08-04

## **Global response**
We sincerely thank all reviewers for their efforts in reviewing our paper and providing invaluable feedback.

This global response reports the results of two additional experiments. While these were primarily conducted to address the comments by Reviewer vPup, notably, Figure 1 in the attached PDF deserves mention here. We thank Reviewer vPup for providing comments that inspired these interesting experiments.

### **Additional experiment 1 (Figure 1): higher noise levels**
We investigated what would happen if we made the training datasets more challenging for our methods to learn projection matrices. We used the synthetic datasets described in Section 6, namely, Packing, MaxFlow, and MinCostFlow, and fixed the dimensionality $k$ of projected LPs to $20$. An important difference from the original experiments in Section 6 is the increased noise level $\omega$, which perturbs LP inputs through multiplication by $1+\omega$. Here, we draw $\omega$ from a uniform distribution over $[0, \bar\omega]$ with the upper bound $\bar\omega$ ranging from $0.0$ to $2.0$ in increments of $0.2$; originally, $\bar\omega$ was fixed at $0.1$. The larger $\bar\omega$ is, the less consistent the tendencies in the input LPs become, making it more challenging to learn effective projection matrices with our PCA- and SGA-based methods.

### **Results**
Figure 1 in the attached PDF presents the objective ratio (i.e., objective values divided by true optimal values computed with "Full") achieved by each method for Packing, MaxFlow, and MinCostFlow datasets.

**Packing.** The performance of our data-driven methods (PCA and SGA) worsens as $\bar\omega$ increases, as expected. While they exhibit a clear advantage over the random-projection baseline (ColRand) at small $\bar\omega$ values, they behave similarly to ColRand when $\bar\omega=2.0$.

**MaxFlow and MinCostFlow.** In contrast to the Packing case, our data-driven methods, particularly SGA, performed well even with high noise levels, while the performance of ColRand remained poor. We guess the success of data-driven methods is due to the fixed graph topology, which creates consistent tendencies across LP instances despite varying edge capacities and costs. Note that fixed graph topologies are ubiquitous in practice; in daily transportation planning, the topology of traffic networks is fixed, while the edge capacities and costs may fluctuate due to congestion. The results highlight the potential benefits of our data-driven projection methods in such applications, newly discovered through this additional experiment.

### **Additional experiment 2 (Figure 2): visualizing the theoretical bound**
We use this experimental result primarily to address the first weakness comment by Reviewer vPup, and thus we detail the experimental settings there. Our purpose here is to describe that the theoretical bound on generalization errors can be non-vacuous when $N$, the dataset size, is sufficiently large relative to problem-size parameters such as $k$ and $n$. While the theoretical bound presented in Figure 2 may appear loose compared with actual generalization errors (approximated by averaging over $20,000$ instances), the bound converges to zero as $N$ increases, offering a meaningful bound on generalization errors when $N$ is large.

Due to constraints on time and computational resources, the above experiments are limited to synthetic datasets (with smaller sizes in the second experiment). Nevertheless, we believe these results have sufficient implications regarding the behavior of our learning methods (PCA and SGA) on more challenging instances and the connection between theory and practice.

---

### Decision · Program_Chairs · 2024-09-25

**Decision:**

Accept (poster)

**Comment:**

This paper presents a data-driven approach to reducing the dimensionality of linear programming (LP) problems by learning projection matrices, which accelerates the solution of high-dimensional LPs. The authors introduce a new generalization bound and propose two specific methods for learning projection matrices using PCA and gradient-based techniques. Experimental results demonstrate that the data-driven projection methods outperform random projection methods across multiple scenarios.

The paper is well-written and clearly structured. Overall, it makes significant contributions both theoretically and practically. While there are concerns that the theoretical results might be too loose in certain experimental settings, the authors have provided additional experiments that show the effectiveness of the generalization bound under specific conditions.